# New Aspects Regarding the Fluorescence Spectra of Melanin and Neuromelanin in Pigmented Human Tissue Concerning Hypoxia

**DOI:** 10.3390/ijms25158457

**Published:** 2024-08-02

**Authors:** Dieter Leupold, Susanne Buder, Lutz Pfeifer, Lukasz Szyc, Peter Riederer, Sabrina Strobel, Camelia-Maria Monoranu

**Affiliations:** 1LTB Lasertechnik Berlin GmbH, 12489 Berlin, Germany; dieter.e.leupold@web.de (D.L.); lutz.pfeifer@ltb-berlin.de (L.P.); 2Clinic for Dermatology and Venerology, Vivantes Klinikum Neukölln, 12351 Berlin, Germany; susanne.buder@vivantes.de; 3Magnosco GmbH, 12489 Berlin, Germany; lukasz.szyc@magnosco.com; 4Department and Research Unit of Psychiatry, University of Southern Denmark, 5230 Odense, Denmark; peter.riederer@uni-wuerzburg.de; 5Center of Mental Health, Department of Psychiatry, Psychosomatics and Psychotherapy, University Hospital Wuerzburg, 97080 Wuerzburg, Germany; 6Institute of Pathology, Department of Neuropathology, University of Wuerzburg, Comprehensive Cancer Center (CCC) Mainfranken Wuerzburg, 97080 Wuerzburg, Germany; sabrina.strobel@uni-wuerzburg.de

**Keywords:** melanin, neuromelanin, melanoma, Parkinson’s, hypoxia, fluorescence

## Abstract

Melanin is a crucial pigment in melanomagenesis. Its fluorescence in human tissue is exceedingly weak but can be detected through advanced laser spectroscopy techniques. The spectral profile of melanin fluorescence distinctively varies among melanocytes, nevomelanocytes, and melanoma cells, with melanoma cells exhibiting a notably “red” fluorescence spectrum. This characteristic enables the diagnosis of melanoma both in vivo and in histological samples. Neuromelanin, a brain pigment akin to melanin, shares similar fluorescence properties. Its fluorescence can also be quantified with high spectral resolution using the same laser spectroscopic methods. Documented fluorescence spectra of neuromelanin in histological samples from the substantia nigra substantiate these findings. Our research reveals that the spectral behavior of neuromelanin fluorescence mirrors that of melanin in melanomas. This indicates that the typical red fluorescence is likely influenced by the microenvironment around (neuro)melanin, rather than by direct pigment interactions. Our ongoing studies aim to further explore this distinctive “red” fluorescence. We have observed this red fluorescence spectrum in post-mortem measurements of melanin in benign nevus. The characteristic red spectrum is also evident here (unlike the benign nevus in vivo), suggesting that hypoxia may contribute to this phenomenon. Given the central role of hypoxia in both melanoma development and treatment, as well as in fundamental Parkinson’s disease mechanisms, this study discusses strategies aimed at reinforcing the hypothesis that red fluorescence from (neuro)melanin serves as an indicator of hypoxia.

## 1. Introduction

### 1.1. Melanin Fluorescence of Human Tissue

Melanin is the primary pigment in human skin and hair and is also present in the eyes. The fluorescence of melanin is exceptionally weak and its spectral distribution in organic solutions exhibits unique characteristics when compared to the typical fluorescence of other organic molecules with pi-electron systems (such as clouds of delocalized electrons from, e.g., carbon 2p orbitals, which define specific molecular absorption and fluorescence properties) [1,2]. This distinctive behavior is also observed in eumelanin and pheomelanin, the two fundamental subtypes of melanin [1]. Additionally, the fluorescence of melanin in human skin tissue cannot be measured using conventional UV excitation (single-photon excitation) due to the significantly stronger fluorescence emitted by other endogenous fluorophores such as keratin, NAD(P)H, and flavins.

However, when the fluorescence of pigmented skin tissue is generated in vivo or in histological preparation as formalin-fixed and paraffin-embedded tissue (FFPE) using two-photon excitation, melanin fluorescence becomes visible. In contrast to the other endogenous fluorophores, melanin shows absorption in the long-wave red spectral range, so it can be excited with a low-energy (“red”) photon to an excited state and from there with a second photon into a higher excited state, from which fluorescence occurs. The other endogenous fluorophores cannot be excited by this step-by-step absorption, so melanin fluorescence appears selectively [1]. This melanin fluorescence exhibits a remarkable spectral diversity. There are characteristic differences depending on whether it originates from normally pigmented skin, from a benign or a dysplastic melanocytic nevus, or from a melanoma (see Figure 1).

These different in vivo fluorescence spectra are identical to those of the fresh excidate, which was fixed directly in formalin as a histological preparation and processed in paraffin. Measurements on such formalin excidates before paraffin preparation show the same four spectral types [2]. This spectral diversity makes melanin fluorescence excited by two photons an objective new method for melanoma diagnosis for the Fitzpatrick skin types 1 to 3 [3,4].

The molecular causes of these spectral differences in melanin fluorescence remain largely unexplained. Changes in the local microenvironment of the melanin in the melanosomes have been postulated [4].

### 1.2. Neuromelanin Fluorescence of the Human Brain Tissue

Neuromelanin has a basic structure similar to melanin [5]. It is found in particular in the substantia nigra, pars compacta of the midbrain. Unlike melanin, there are no hints in the literature on neuromelanin fluorescence in solutions. Histological samples of the substantia nigra show clear spectral differences in the two-photon-excited fluorescence between measurements in the pigmented and non-pigmented areas of the adjacent parenchyma containing predominantly glial elements [5].

### 1.3. Melanin Fluorescence—Measurements on Preparations Obtained Postmortally

The described fluorescence measurements of melanin and neuromelanin differ in that the former were performed in the in vivo state or in the formalin-fixed ex vivo state (histological preparation), whereas the neuromelanin studies were performed on post-mortem samples. As a bridge between the two, we report here about melanin fluorescence measurements on postmortem tissue.

## 2. Results and Discussion

Figure 2 shows a representative measurement result of the pigmented skin lesion removed post-mortem and histologically characterized as a benign melanocytic nevus. Red areas indicate red melanin fluorescence according to Figure 1a. According to current knowledge of in vivo measurements, they would therefore characterize micro-areas of a malignant melanoma. Their frequency and distribution together with that of the yellow areas (characterized as dysplastic micro-areas with spectra according to Figure 1b) would lead to the automatic overall diagnosis of melanoma. In contrast, a benign melanocytic nevus in vivo or fixed ex vivo (histological preparation) would show predominantly green areas (spectra according to Figure 1c). However, this appearance in Figure 2 belongs to a histologically benign melanocytic nevus in the post-mortem state, i.e., in a state of oxygen deficiency. This suggests that hypoxia in the microenvironment of the melanocytic melanin causes the red fluorescence shown in Figure 1a.

On the other hand, since malignant melanomas in vivo are characterized by this red melanin fluorescence (it is their reliable indicator both in vivo and in the fixed ex vivo state in the histological preparation), it can be assumed that hypoxia is generally present in the red fluorescent areas of a melanoma.

### A Bridge to Neuromelanin Fluorescence of the Substantia Nigra

There is ample evidence that patients with Parkinson’s disease (PD) are at risk for developing melanoma and vice versa [6,7,8], indicating common genetic components [9]. 129Ser-phosphorylated α-synuclein, a pathological marker for PD, has been shown to be expressed in cutaneous malignant melanoma but not in normal skin [10,11]. Hypoxia has also been discussed as pathological hallmark of PD. Indeed, hypoxia may trigger PD [12,13,14], but has also been suggested to improve symptomatology in clinical settings [15,16]. As such, it is of interest that Hypoxia-Inducible Factor (HIF) 1a has been shown to be substantially decreased in the substantia nigra (SN) of PD [17,18]. HIF isoforms regulate cellular oxygen concentration important for adaptive mechanisms, when oxygen and reactive oxygen species homeostasis is unbalanced [19,20,21]. Metabolic energy decline has been found to be of critical pathological evidence in PD [22].

Of note, an interaction of HIF with α-synuclein [23,24,25,26] and iron [27,28,29] has been demonstrated. And α-synuclein as well as iron are bound to neuromelanin [30,31,32,33,34]. To conclude, there is abundant evidence that hypoxia plays an important factor in the multiple pathological interactions underlying PD. In addition, hypoxia as a result of comatous stages in the final phase of diseases including PD may have implications to modify outcomes of analytical measurements, such as those reported here.

The previously published two-photon fluorescence spectra of neuromelanin in the substantia nigra were obtained from histological preparations [5]. In contrast to the histological samples of pigmented skin obtained from the living patients, the brain preparations originate from the post-mortem state. The spectral shape of these neuromelanin spectra is the same as that shown in Figure 1a. Figure 3 shows a comparison of the standardized red melanin fluorescence spectrum with that of neuromelanin in histological preparations of substantia nigra (*post-mortem*), which match.

This suggests the conclusion that hypoxia in the (neuro)melanin microenvironment of the post-mortem brain is the dominant cause of the red spectral form, and that the red melanin fluorescence that occurs in the skin of the living/in vivo during malignant melanocytic degeneration is apparently a consequence of local hypoxia. This is also supported by the occurrence of red melanin fluorescence in pigmented basal cell carcinoma [3].

Neuromelanin fluorescence spectra from the SN of patients with PD obtained from histological preparations are also shown in a previous publication [5]. They are identical to those of non-Parkinsonian subjects. This suggests that in both cases, the influence of hypoxia on the spectra is dominant. The interesting question of whether PD is reflected in the neuromelanin fluorescence spectra (whether it could be a sensitive indicator) is therefore still unanswered; neuromelanin fluorescence spectra in vivo (or close to the in vivo state) would be required for this. In a previous paper [5], it was concluded from this spectral identity that the neuromelanin structure/microenvironment in the substantia nigra of Parkinsonians does not differ from that of healthy subjects. This must be regarded as unproven for the in vivo state on the basis of the available studies; the statement made in [5] only applies to the post-mortem state. On the other hand, this result confirms also that there is no indication of this disease in the spectral course of the melanin fluorescence of pigmented normal skin or benign and dysplastic nevi of Parkinsonians in vivo.

## 3. Material and Methods

The following results were obtained from a histological sample of a pigmented lesion excised post-mortem from the forearm of a 66-year-old man. The sample was obtained with the consent of next of kin according to the guidelines of the NIH Guide for the Care and Use of laboratory human tissue and were approved by the local ethics committee of the University of Wuerzburg (internal application number 99/11). The lesion, clinically classified as a benign melanocytic nevus, was subjected to histopathological examination which confirmed the diagnosis. A second histological examination was also performed by an independent pathologist which confirmed the previous result. The fluorescence measurement was performed at Lasertechnik Berlin GmbH (LTB) on the direct sample and after removal of a 2 µm layer to avoid surface artefacts during the measurement.

### Principle of the Measurement

The fluorescence of these two samples was measured using the method known as dermatofluoroscopy, described in detail in the literature [4,5] (LTB Lasertechnik Berlin GmbH, Berlin, Germany). In brief, the fluorescence is excited by 800 nm/1 ns pulses. Each measuring pulse analyzes a tissue area with a diameter of 30 µm. Spectrally and spatially resolved signals (up to several hundreds of spectra per lesion, depending on the scanned area) are detected with a cooled CCD (charge-coupled device) camera (in the spectral range 380–780 nm) and automatically assigned to one of the four types of Figure 1. (Spectra that could not be reliably assigned to any type are eliminated). The measurement result is displayed as a color image on the measurement sample (see Figure 2).

The measurement data analysis also provides the assignment of the sample as a benign or dysplastic nevus or as a melanoma on the basis of a score (a measure, based on the number and size of measurement point clusters of spectra of type Figure 1a; see [3]). This measurement data analysis was successfully evaluated in a multicenter clinical study [3]. An analogous application of this method for melanoma histological diagnosis has been documented.

## 4. Conclusions

The relationship between hypoxia and malignization, e.g., the transformation of a nevus into a melanoma, is the subject of increasing research activity [35,36,37]. For example, in malignant melanoma of the skin and uveal melanoma, the possibilities and effects of targeted modulation of hypoxia in the tumor microenvironment are being investigated [36]. For example, active reduction in hypoxia may improve photodynamic tumor therapy (PDT), which is based on the action of toxic oxygen radicals (ROS). The possibility of high-resolution hypoxia detection in pigmented tissues described here may be of considerable benefit to current cancer research.

A link with hypoxia is also currently being investigated in the progress of PD pathology [38,39]. Therefore, the question raised in the previous study [5] of a reflection of Parkinson’s degeneration in the neuromelanin fluorescence of the substantia nigra is relevant, but would require a new approach, namely (an approximation of) measurements in vivo.

## Figures and Tables

**Figure 1 ijms-25-08457-f001:**
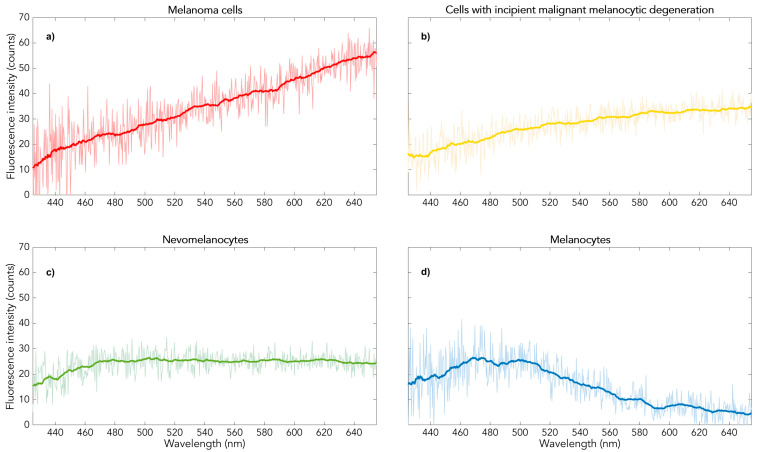
Two-photon-excited melanin fluorescence spectra of pigmented skin from fair-skinned people (Fitzpatrick type 1–3) in vivo or in the fixed in vivo state FFPE: (**a**) (red): melanoma, (**b**) (yellow): dysplastic nevus, (**c**) (green): benign nevus, (**d**) (blue): normally pigmented skin.

**Figure 2 ijms-25-08457-f002:**
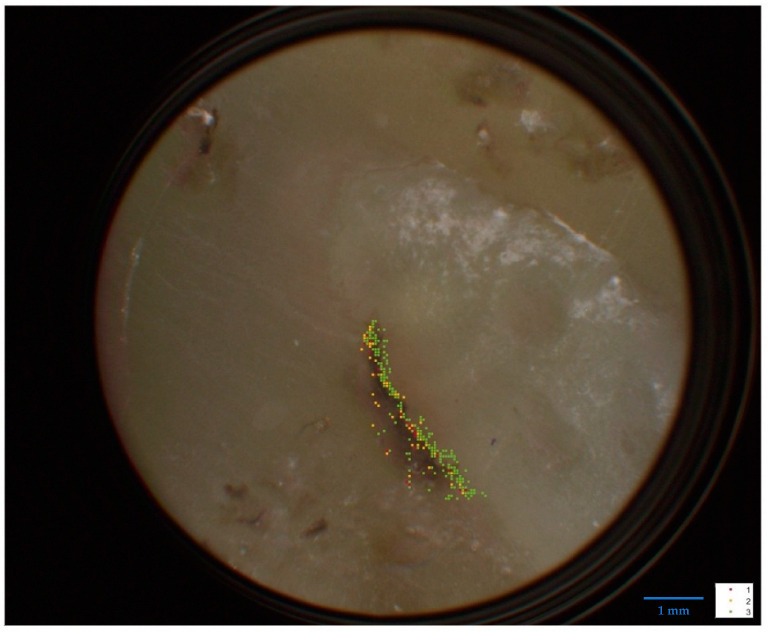
DermatoHistoFluoroscopy—analysis of an FFPE specimen of a benign nevus (histopathological findings) from a deceased person. The colors here characterize the spectral progression as shown in Figure 1, but the tissue classification there is not transferable (here post-mortem condition, whereas the spectra of Figure 1 were measured on tissue in vivo or in the fixed in vivo state FFPE).

**Figure 3 ijms-25-08457-f003:**
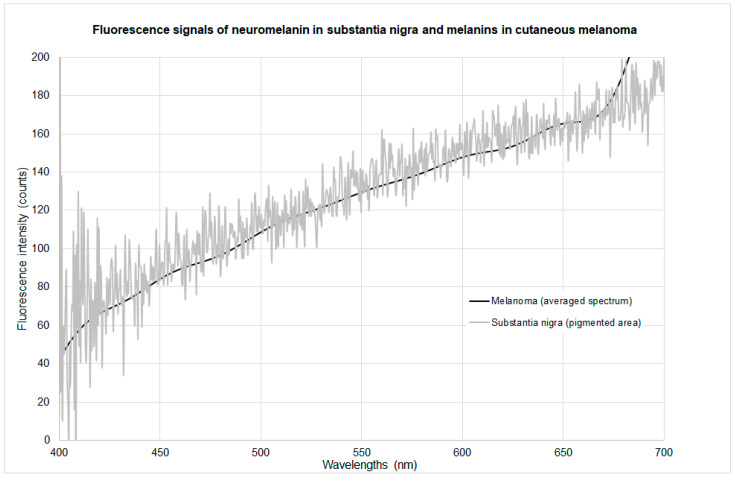
Spectra of two-photon-excited melanin and neuromelanin fluorescence in hypoxic microenvironment.

## Data Availability

The original contributions presented in the study are included in the article, further inquiries can be directed to the corresponding author.

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
