# Peer review of "New Aspects Regarding the Fluorescence Spectra of Melanin and Neuromelanin in Pigmented Human Tissue Concerning Hypoxia"

_ijms, 2024, doi:10.3390/ijms25158457_

Round 1

Reviewer 1 Report

Comments and Suggestions for Authors

Main comments

The authors present results on fluorescence scanning of one melanin sample of post-mortem begnin nevi and discuss results about neuromelanin obtained previously. They found that both types of samples present characteristics of melanoma and concluded that it may be due to hypoxia. The paper seems to be a quick paper to get out a hypothesis, however there is no proof advanced that hypoxia is the driver of the red fluorescence in the different post-mortem samples and melanoma, even though there is a good discussion on the subject. The link between neuromelanin and the nevus sample is clear in the results part but not in the abstract and not in the last paragraph of the introduction.

The paper would benefit of a rewriting and a bigger sample size and heathy treated to get hypoxia and not would allow to conclude on the effect of hypoxia.

In the methods, it is not clear the number of sample. The first paragraph indicates n=1 and the second n=2?

Minor comments

Abstract:

L21 repetition, maybe one sentence with melanoma and neuromelanin methods, so that the abstract is more fluent. What is the question of the paper? What are the results?

Introduction

L38: what is the “pi-electron system”

L45: the method could be developed so that we understand what is done in this article.

L52-4: Are the melanocytes cell lines? Primary cells? Idem for neuromelanocytes.

L66 “neuromelanin has 66 not been shown fluorescence in solvents.” It could be written: neuromelanin does not fluoresce in solvent.

L67-69 “Histological samples of the substantia nigra 67 show clear spectral differences in the two-photon excited fluorescence between measure-68 ments in the pigmented and non-pigmented areas of the adjacent parenchyma containing 69 predominantly glial elements.” Do you have a reference?

L74: it is not clear to what the authors refer to? 

L75: explain better what the authors want to show in this article, it is not clear from this sentence.

Methods

The first paragraph of the methods looks like the beginning of a result paragraph.

L90: It is not clear how the experiment was performed. How many measurements? How many discarded? On which area?

L96: what does it mean “not transferable”

L98: on which score?

L108. A punctuation is missing.

L112 and 114: the hypoxia can be suggested but it cannot be concluded. An experiment should be done to show it to conclude.

L121: 129 up case.

L125: SN? Definitition

L146: “demonstrate” to strong.

Ref 4-9  this two references are the same paper

Comments on the Quality of English Language

The are some misspelling and editing mistakes to be corrected, but in general it is fine.

Author Response

Response to Reviewer 1 Comments

We thank you for your expert comments and the opportunity to respond to the criticisms of our manuscript, modifications are marked in the text as follows: parts highlighted in gray to be removed; paragraphs highlighted in yellow are newly added.

Comments and Suggestions for Authors

Main comments

The authors present results on fluorescence scanning of one melanin sample of post-mortem begnin nevi and discuss results about neuromelanin obtained previously. They found that both types of samples present characteristics of melanoma and concluded that it may be due to hypoxia. The paper seems to be a quick paper to get out a hypothesis, however there is no proof advanced that hypoxia is the driver of the red fluorescence in the different post-mortem samples and melanoma, even though there is a good discussion on the subject. The link between neuromelanin and the nevus sample is clear in the results part but not in the abstract and not in the last paragraph of the introduction.

Answer: See changes in the title, in the abstract as well in the text (highlighted in yellow)

The paper would benefit of a rewriting and a bigger sample size and heathy treated to get hypoxia and not would allow to conclude on the effect of hypoxia.  

Answer: The special measuring device for melanin fluorescence is currently not available to the authors, a bigger sample size is planned for a next study.

In the methods, it is not clear the number of sample. The first paragraph indicates n=1 and the second n=2?

Answer: One histological sample was measured at two different cutting planes (highlighted in yellow in the text)

Minor comments

Abstract:

L21 repetition, maybe one sentence with melanoma and neuromelanin methods, so that the abstract is more fluent. What is the question of the paper? What are the results?

Answer: We added to the manuscript: The present works aims at further characterizing this “red” fluorescence…. And thus points to hypoxia as at least an accompanying cause.

We also added to the abstract: In light of the central role of hypoxia in melanoma development and therapy as well as basic Parkinson mechanisms, strategies to further secure the working hypothesis of red (neuro)melanin fluorescence as a hypoxia indicator are discussed.

Introduction

L38: what is the “pi-electron system”:

Answer: Explanation was added to the manuscript:

(clouds of delocalized electrons from e.g. carbon 2p orbitals of organic molecules, which determine the specific molecular absorption and fluorescence properties). The same applies to eumelanin and pheomelanin, the two basic subtypes of melanin (1,2).

L45: the method could be developed so that we understand what is done in this article.

Answer: was added to the manuscript:

In contrast to the other endogenous fluorophores, melanin shows absorption in the long-wave red spectral range, so it can be excited with a low-energy ("red") photon to an excited state and from there with a second photon into a higher excited state from which fluorescence occurs. The other endogenous fluorophores cannot be excited by this step-by-step absorption, so melanin fluorescence appears selectively.

L52-4: Are the melanocytes cell lines? Primary cells? Idem for neuromelanocytes.

Answer:  thank you for this question; the previous figure caption was misleading; it has now been corrected

L66 “neuromelanin has 66 not been shown fluorescence in solvents.” It could be written: neuromelanin does not fluoresce in solvent.

Answer: Correct is: there are no hints in literature on neuromelanin fluorescence in solutions. (was added to the manuscript)

 L67-69 “Histological samples of the substantia nigra 67 show clear spectral differences in the two-photon excited fluorescence between measure-68 ments in the pigmented and non-pigmented areas of the adjacent parenchyma containing 69 predominantly glial elements.” Do you have a reference?

Answer: reference [11]

L74: it is not clear to what the authors refer to? 

L75: explain better what the authors want to show in this article, it is not clear from this sentence.

Answer L74-75: the sentence was modified: As a bridge between the two, we report here about melanin fluorescence measurements on postmortem tissue.

Methods

The first paragraph of the methods looks like the beginning of a result paragraph.

Answer: This first paragraph characterizes the material

L90: It is not clear how the experiment was performed. How many measurements?

Answer: Several hundreds, see text. How many discarded? Less than 1% On which area? 30 µm in diameter per pulse, see text. The total area is shown in Fig.2 (was added to the manuscript)

L96: what does it mean “not transferable”

Answer: The spectra of Figure 1 were measured on tissue in vivo or in the fixed in vivo state FFPE, those of Figure 2 on tissue in the post mortem state

L98: on which score?

Answer: The score is a measure based on the number and size of measurement point clusters of spectra of type 1a in Fig.1

L108. A punctuation is missing.

Answer: was added to the manuscript

L112 and 114: the hypoxia can be suggested but it cannot be concluded.

Answer: Accepted, see text: An experiment should be done to show it to conclude.

L121: 129 up case

Answer: sorry, we don't understand what is meant here?

L125: SN? Definitition

Answer: we added “substantia nigra” to the text before of the first abreviation

L146: “demonstrate” to strong.

Answer: was changes to “suggests the conclusion” instead of “demonstrate”

Ref 4-9  this two references are the same paper

Answer: Thank you, was corrected

Comments on the Quality of English Language

The are some misspelling and editing mistakes to be corrected, but in general it is fine.

Reviewer 2 Report

Comments and Suggestions for Authors

The work is interesting and deserves publication, but the authors should consider the following comments:

-There are two main types of melanins: eumelanins and pheomelanins. Has this fluorescence technique been applied to these types of melanin?

-The enzyme that initiates melanin biosynthesis is tyrosinase. Can you tell if it interferes with the fluorescence measurements?

Author Response

Response to Reviewer 2 Comments

We thank you for your expert comments and the opportunity to respond to the criticisms of our manuscript, modifications are marked in the text as follows: parts highlighted in gray to be removed; paragraphs highlighted in yellow are newly added.

The work is interesting and deserves publication, but the authors should consider the following comments:

-There are two main types of melanins: eumelanins and pheomelanins. Has this fluorescence technique been applied to these types of melanin?

Answer: was added to the manuscript: “The same applies to eumelanin and pheomelanin, the two basic subtypes of melanin (1,2)

Additional answer (only for the Reviewer):

Using the two-photon excitation method, extensive fluorescence studies were carried out on eumelanin and pheomelanin, each in different solvents ([1-3], see also the full reference list in [6]). Originally, the hypothesis was also pursued that the ratio of eumelanin to pheomelanin could determine the red shift of tissue fluorescence in malignant melanocytic degeneration. This has not been confirmed. Rather, the following applies:

In conclusion, the increasing redshift of the melanin fluorescence in the

process of malignant transformation from normal melanocytes to melanoma cells is caused

by changes in the local microenvironment of the melanin in the melanosomes. In the case

of melanoma in particular, this specific microenvironment exists in all subtypes regardless of their genetic and anatomical characteristics and their eumelanin/pheomelanin ratio.

-The enzyme that initiates melanin biosynthesis is tyrosinase. Can you tell if it interferes with the fluorescence measurements?

Answer:  The role of tyrosinase in the synthesis of substantia nigra neuromelanin is still a question of discussion and research: (Nagatsu et al. 2023). We don’t added to the manuscript.

Nagatsu T, Nakashima A, Watanabe H, Ito S, Wakamatsu K, Zucca FA, Zecca L, Youdim M, Wulf M, Riederer P, Dijkstra JM (2023) The role of tyrosine hydroxylase as a key player in neuromelanin synthesis and the association of neuromelanin with Parkinson's disease. J Neural Transm (Vienna) 130 (5):611-625. doi:10.1007/s00702-023-02617-6

Round 2

Reviewer 1 Report

Comments and Suggestions for Authors

Main

The authors improved the manuscript. They present the reasons for red fluorescence detected in neuromelanin samples, by comparing this result to results obtained from fluorescence of a post-mortem benign nevus and hypothesised that the hypoxia could be the cause. 

The abstract and the introduction need some work to be more fluent, there are missing link between sentences.

Abstract

L27-31: the sentences should be rephrased to have a more fluent text.

Intro

L41 “colouring pigment”: a pigment is by definition something with a colour.

L42: A reference is missing

L42-46: The flow is not easy to follow: “Its spectral” = melanin?, then “the same applies to eumelanin and pheomelanin” = melanin?.

L46: the authors mean: the measured fluorescence?

Comments on the Quality of English Language

The English should be more fluent in the abstract and introduction

Author Response

A letter for the reviewer is attached
